# Nutritional Composition and Bioactive Compounds in Three Different Parts of Mango Fruit

**DOI:** 10.3390/ijerph18020741

**Published:** 2021-01-16

**Authors:** Veeranjaneya Reddy Lebaka, Young-Jung Wee, Weibing Ye, Mallikarjuna Korivi

**Affiliations:** 1Department of Microbiology, Yogi Vemana University, Kadapa 516003, India; lvereddy@gmail.com; 2Department of Food Science and Technology, Yeungnam University, Gyeongsan 38541, Korea; yjwee@ynu.ac.kr; 3Exercise and Metabolism Research Center, College of Physical Education and Health Sciences, Zhejiang Normal University, Jinhua 321004, China

**Keywords:** tropical fruit, nutraceutical composition, mango byproducts, health benefits

## Abstract

Mango (*Mangifera indica* L.), known as the king of fruits, has an attractive taste and fragrance and high nutritional value. Mango is commercially important in India, where ~55% of the global crop is produced. The fruit has three main parts: pulp, peel, and kernel. The pulp is the most-consumed part, while the peel and kernel are usually discarded. Mango pulp is a source of a variety of reducing sugars, amino acids, aromatic compounds, and functional compounds, such as pectin, vitamins, anthocyanins, and polyphenols. Mango processing generates peels and kernels as bio-wastes, though they also have nutraceutical significance. Functional compounds in the peel, including protocatechuic acids, mangiferin and β-carotene are known for their antimicrobial, anti-diabetic, anti-inflammatory, and anti-carcinogenic properties. The mango kernel has higher antioxidant and polyphenolic contents than the pulp and peel and is used for oil extraction; it’s possible usage in combination with corn and wheat flour in preparing nutraceuticals is being increasingly emphasized. This review aims to provide nutraceutical and pharmacological information on all three parts of mango to help understand the defense mechanisms of its functional constituents, and the appropriate use of mangoes to enhance our nutrition and health.

## 1. Introduction

Mango has been cultivated in India for more than 4000 years. Among the other tropical fruits, mango is the most popular dessert worldwide. Mango belongs to the Anacardiaceae family, which includes a number of deadly poisonous plants. It has a delicious taste (delightfully blended sweetness and acidity) and aroma, and high nutritional value. The production of mango continues to rank it as the predominant tropical fruit in the 21st century [1]. The global mango production reached 51 million tons in 2019. Most of the mango crop is produced in Asia, especially India, where production has reached 22 million tons per year, remaining the top exporter in the world [1,2]. Due to the optimal geographical and climatic conditions, the state of Andhra Pradesh is ideal for fruit and vegetable cultivation. Mango, banana, papaya, coconut, pomegranate, sweet orange, grape, and cashew are the principal fruit crops cultivated in Andhra Pradesh. Mango is cultivated on about 0.37 million hectares in Andhra Pradesh, with a production of 4.69 million tons, which is the highest level of production in the entire country (a 20% share). Due to the rising international popularity of fresh and processed mango products, in 2015–2016, India shipped about 36,000 tons of mangoes and 129,000 tons of pulp overseas [3]. Mango is cultivated in 85 countries worldwide; Asian countries, including India, China, Thailand, and Indonesia provide for 80% of the total world production (Figure 1a).

Mango is economically important, being the third-largest agricultural product in India. The cultivation of mango in India covers around 2.26 million acres, which is 40% of the total area used for Indian fruit production (Figure 1b). Over 30 varieties of mangoes are commercially available in India. The huge diversity in Indian mango cultivation is due to the large number of cultivars (the country is home to approximately 1000 varieties, only a few of which are commercially cultivated) and wild varieties. Apart from *Mangifera indica*, many other species such as *M. khasiana*, *M. andamanica*, *M. camptosperma* and *M. sylvatica* have been reported in India [4].

Fruit plays an important role in the world’s economy and food security from various perspectives [5]. However, economic success or failure of commercial mango production primarily relies on local weather and climate changes. A recent study indicated that the frequency and severity of extreme weather events have affected mango production in Central Queensland, Australia [6]. Although mango trees are adapted to dry weather, weather factors like floods, rainfall, humidity, and temperature can influence tree growth, flowering, fruit growth, the color and size of fruit, and farmer income [6,7,8]. Temperature has a dominant influence on the appearance, quality, and taste of mango fruit. The increasing temperatures are offering opportunities for mango production in new areas. The mangoes grown in the northern states of India and Thailand require longer to mature than those in the central and southern states [7]. High temperature has positive effects on mango fruit growth and maturation. The estimated duration of mango fruit development decreased by 12–16 days in Australia due to the rise in winter temperature (1.5 °C) over the last 45 years [9]. In addition, high-temperature-induced stress can trigger the synthesis of secondary metabolites, which are popular due to their nutritional and medicinal values [10]. 

Mango contains a blend of sugar (16–18% *w*/*v*) and acids and high amounts of antioxidants (ascorbic acid) and polyphenols (carotene, as vitamin A). The principal carbohydrates that are used are different in green unripe and matured ripe mango [4]. Starch is the principal carbohydrate in green mango; during maturation, it converts to reducing sugars (sucrose, glucose, and fructose). Along with these carbohydrates, small quantities of cellulose, hemicellulose, and pectin are present in ripe mango [4]. Unripe mango tastes sour because of the presence of different acids, such as citric acid, malic acid, oxalic acid, succinic, and other organic acids, whereas the sweet taste of the ripened fruit is due to the blending of reducing sugar and the main acid source, malic acid [11]. High concentrations of β-carotene and other phytochemicals in mangoes can prevent leukemia and progression of prostate, breast, and colon cancers [12,13,14,15,16]. Mangoes can be differentiated in to three parts: pulp (mesocarp), peel (epicarp), and seed kernel (endocarp), as presented in Figure 2.

## 2. Mango Pulp

Pulp is the main and directly consumable part of the mango. Mango pulp accounts for 50s to 60% of the weight of the total fruit, and is used to prepare various products like juice, jam, puree, and nectar. Pulp is the source of many nutritional and functional compounds. Mango pulp is used in dairy and beverage industries as a flavoring agent and in neonatal food formulations. The market for mango-related products is steadily increasing, with an annual growth of 5% [1,3]. The Totapuri cultivar is used extensively for pulp production because of its high pulp-yielding rate. During the processing, 60% to 75% of pulp is produced and the remaining 25% to 40% is generated as a waste byproduct [17]. The pulp-processing units usually collect mangoes from the designated market yards. Around 10–12 tons of processing byproducts (peel and stones) are generated per day from a plant that processes 40 tons of Totapuri mangoes [18,19,20]. The factories dump this processing residue in open fields, the accumulation of which may cause environmental issues if not managed. This residue is usually used by surrounding people or farmers as feed for their cattle [21,22]. The World Health Organization (WHO) recommends the daily consumption of 400 g of fruits and vegetables, as their nutrients could prevent chronic diseases, such as heart diseases, cancer, diabetes, and obesity [23]. Therefore, demand is increasing for mango as a functional food. Bioactive compounds provide color to the fruits and vegetables; therefore, more quantities of colorful plant products should be consumed to promote health [24].

### 2.1. Nutritional Composition

The nutritional composition of mango pulp mainly depends on the type/variety of the mango, the locality and climatic conditions of its production region, and the maturity of the fruit [25,26,27,28]. The nutritional value of mango is shown in Table 1. Mango contains a variety of macro- and micronutrients. In terms of macronutrients, the mango pulp contains carbohydrates (16–18%), proteins, amino acids, lipids, organic acids, as well as dietary fiber [29]. The pulp is also a good source of micronutrients, including trace elements such as calcium, phosphorus, iron, and vitamins (vitamins C and A). Consumption of mango pulp provides high energy: 60–190 Kcal from 100 g of fresh pulp. Along with the above essential nutritional elements, mango pulp contains 75–85% water (Table 1). 

#### 2.1.1. Macronutrients

Mango contains different types of carbohydrates based on the stage of maturity. Ripe mango is a rich a source of sugars (fructose, glucose, sucrose), whereas unripe mango is a source of starch and pectin. During the ripening process, starch is converted to fructose and glucose. The increase in mono- and disaccharide concentrations upon maturation is observed in many varieties, including Alphonso [30,31], Deshahari [32], and Tommy Atkins [33]. Pectin is a gelling sugar and is important for preserving the firmness of the fruit; its levels decrease during the ripening, making the flesh sweeter and smoother [26,34]. Sucrose, fructose, and glucose (in decreasing order of their concentration) are the principal sugars present in mature and ripe mango [26,30,35].

Compared to carbohydrates, the protein content in mango fruit is low (0.5–5.5%) (Table 1). Depending on the region of cultivation, mangoes have different protein contents. For example, the highest protein content is found in mangoes produced in Peru, and the lowest protein is found in those from Columbia. The content of constituent amino acids varies with the maturity level, region, and species of the fruit [36]. The usually occurring amino acids in the pulp are leucine (6.9 g), lysine (4.3 g), methionine (1.2 g), threonine (3.4 g), valine (5.8 g), arginine (7.3 g), glutamic acid (18.2 g), glycine (4.0 g) and proline (3.5 g) per 100 g protein [35]. 

#### 2.1.2. Minerals and Vitamins (Micronutrients) 

Vitamins and minerals are important micronutrients present in mango pulp. According to the United States Department of Agriculture (USDA) database, mango pulp comprises four different types of water- and fat-soluble vitamins. The concentrations of vitamins A and C are higher compared to those of vitamins B, E, and K (Table 1) [37]. Similar to the sugar content, the organic acid type and content changes as the fruit matures, being also dependent on the locality and species. The quantity of vitamin C ranges from 98 mg to 18 g/kg; thus, consumption of mango on regular basis could fulfill the recommended dietary intake of vitamins C and A [23]. Vitamin C, an antioxidant and immune booster, is required for collagen regeneration, scurvy prevention, and iron absorption [38]. The USDA [39] reported 36.4 mg/100 g of vitamin C, on average, in mango pulp (Table 1). Vitamin A and its metabolites have attracted interest due to their antioxidant activity, vision benefits, immunity, and beneficial effects against cancer and cardiovascular diseases [40]. The concentration of vitamin A ranges from 1000–6000 IU, and the consumption of one fresh fruit (250–300 g) provide 10–12% the recommended daily amount (RDA) of retinol. Mango consumption is thus one of the best ways to prevent vitamin A deficiency [41,42]. Compared to vitamin A, vitamin E and K concentrations are lower in mango pulp. Their amounts also increase as the fruit ripens. Conversely, Indian varieties like Deshahari possess low levels of vitamin E in mature fruit [43]. Fresh mango pulp (100 g) contains roughly 1.3 mg of α-tocopherol, an active form of vitamin E. [44]. An inverse relationship exists between the contents of vitamin C and vitamin E in mango fruit [45]. The vitamin B complex is important for both plant and human metabolism and nutrition. The mango pulp contains all the B complex vitamins except biotin. The B complex vitamin content in mango changes as the fruit matures, ranging from 1.5–2.5 mg/100 g of fresh fruit pulp [26]. Mango pulp is a good source of elemental minerals that are essential for a variety of biochemical reactions. The consumption of mango provides amounts of many micro- and macrominerals such as calcium, sodium, copper, iron, phosphorus, manganese, magnesium, zinc, boron (0.6–10.6 mg/kg), and selenium (Table 1).

**Table 1 ijerph-18-00741-t001:** Nutritional and functional phytochemical composition of mango pulp, peel, and seed kernel.

Compound (Per 100 g)	Pulp	Peel	Seed Kernel
Water (g)	83.46	72.5	9.1
Energy (kcal)	60		327
Carbohydrate, by difference (g)	14.98	28.2	18.2
Protein (g)	0.82	3.6	6.61
Total lipid (fat) (g)	0.38	2.2	9.4
Sugars, total (g)		25	70
Total dietary fiber (g)	1.6	40–72.5	2.8
**Minerals (mg)**			
Calcium (Ca)	11	150	450
Iron (Fe)	0.16	40.6	11.9
Magnesium (Mg)	10	100	100
Phosphorus (P)	14	-	140
Potassium (K)	168	75	365
Sodium (Na)	1	50	150
Zinc (Zn)	0.09	1.74	1.1
Copper (Cu)	0.04–0.32	10.4	-
Selenium (Se)	0–0.6	-	-
**Vitamins**			
Vitamin C (total ascorbic acid, mg)	36.4	18–257	17
Thiamin (mg)	0.028		0.08
Riboflavin (mg)	0.038		0.13
Niacin (mg)	0.669		0.19
Pantothenic acid (mg)	0.119		0.12
Folate, dietary folate equivalents (μg)	43		-
Vitamin A, retinol activity equivalents (μg)	54	100	-
Vitamin E (α-tocopherol, mg)	0.9	0.25–0.59	1.3
Vitamin A (IU)	1082	-	15
Vitamin K (phylloquinone, μg)	4.2		59
Vitamin B12			0.12
**Organic acids**			
Citric acid (%)	0.7		
Mallic acid (%)	0.5		
**Polyphenols**			
Cyanidin (mg)	0.1		
Catechin (mg)	1.7		
Kaempferol (mg)	0.1	3.6	
Myricetin (mg)	0.1		
Proanthocyanidin dimers (mg)	1.8		
Proanthocyanidin trimers (mg)	1.4		
Proanthocyanidin 4–6 mers (mg)	7.2		
Gallic acid	0.69		23–838
Ellagic acid			3–156
Coumarin			12.7
Caffeic acid			7.7
Vanillin			202
Cinnamic acid			11.2
Ferulic acid			10.4
Mangiferin (mg)		169	4.2
Mangiferin gallate		321	
Isomangiferin		13.4	
Isomangiferin gallate		82	
Quercetin	2.2	6.5	
Rhamnetin 3-0 galactoside/glucoside		9.4	
Tannins			20.7
**Flavonoids (catechin equivalent/100 g)**	0.9–9.2	19.91–75.35	
**Anthocyanins**		360–565	
Cyaniding		22.1	
Pelargonidins		22.73	
Delphinidins		18.02	
Malvidins		5.26	
Petunidins		21.6	
Peonidins		24.42	
**Carotenoids** (μg)		3092	
β-carotene	640	1310	
α-carotene	9		
β-cryptoxanthin	10	600	
Lycopene	3		
Lutein and zeaxanthin	23	299	

Sources: Tharanathan et al. [36]; Dar et al. [26]; United States Department of Agriculture (ARS) [35]; Burton-Freeman et al. [18]; Maldonado-Celis et al. [25].

#### 2.1.3. Lipids and Fatty Acids 

The lipid and fatty acid proportion in mango pulp is lower than the protein content and much lower than the carbohydrates. The fatty acid composition of pulp was studied in many mango varieties, and triglycerols were found to be the major constituent along with minor levels of mono- and diglycerols. The total content ranges from 0.8% to 1.36% [46]. Mango pulp contains essential fatty acids, the concentrations of which increase during ripening and stabilize finally at around 1% [47]. The fatty acid concentration is used as an index of the maturity of the mango, with a palmitic–palmitoleic acid ratio of one indicating the fully ripened stage [47,48]. 

#### 2.1.4. Organic Acids 

Mango pulp contains various organic acids including citric acid, malic acid, oxalic acid, succinic acid, ascorbic acid, and tartaric acid. Organic acids are generally weak acids with low molecular weight. The presence of organic acids provides fruits their characteristic tastes and flavors, playing an important role in their organoleptic quality [37]. Among others, citric acid, malic acid, and succinic acid are the major organic acids present in various types of Indian mangoes [49]. 

### 2.2. Phytochemical Composition

The mango pulp is known to have excellent bioactive compounds like carotenoids (pro-vitamin A, 3894 IU/100 g), phenolic acids, polysaccharides, sterols, and alkaloids. Mango is included in the TRAMIL list (a research project on medicinal plant resource in the Caribbean) as it is used by the indigenous people to treat diarrhea, fever, gastritis, and ulcers [50]. 

#### 2.2.1. Phenolic Compounds 

Phenolic compounds are the important secondary metabolite and can be categorized into phenolic acids and polyphenols. These are mainly present in combination with sugar moieties, linked to one or more phenolic groups, or can occur as ester or methyl-ester derivatives. Mango is one of the rich sources of these phenolic compounds. 

##### Phenolic Acids 

Phenolic acids are the important secondary metabolites that help to protect against various diseases and play pivotal role in management of human health. Pulp contains mainly hydroxycinnamic- and hydroxybenzoic-derivative phenolic acids, which are present in free form or conjugated with glucose or quinic acid or both. Gallic, syringic, vanillic, and protocatechuic acids are the main constituents of the hydroxybenzoic acid group, and p-coumaric, ferulic, chlorogenic, and caffeic acids are the main constituents of the hydroxycinnamic acid group [51]. The phenolic acid type and concentrations vary with mango variety, growing location, and maturity stage. In the pulp of most mango varieties, the ferulic acid has the highest concentration (33.75 mg), followed by chlorogenic (0.96–6.20 mg), gallic (0.93–2.98 mg), vanillic (0.57–1.63 mg), protocatechuic (0.77 mg), and caffeic (0.25–0.10 mg) acids per 100 g of fresh fruit weight [28]. Conversely, chlorogenic acid is the main constituent (90%) in Ataulfo mango pulp, followed by gallic (4%), vanillic (30%), and protocatechuic (56%) acids at the final ripe stage.

Hu et al. [52] identified 34 different phenolic acid derivatives, such as gallotannins and quercetin, and, for the first time, identified rosmarinic acid in mango pulp using UPLC-ESI-QTOFMS. Ramirez et al. [53] identified mangiferin, homomangiferin, and dimethyl mangiferin in the pulp of Tommy Atkins and Pica varieties. Mangiferin is a yellow crystal that chemically belongs to the xanthone family. Mangiferin is known for its pharmacological activities, including anticancer, antimicrobial, anti-atherosclerotic, antiallergenic, anti-inflammatory, analgesic, and immunomodulatory activities [26,54,55,56,57].

#### 2.2.2. Pigments 

The maturation stage of the fruit can be determined by changes in the color and texture of the fruit peel and flesh. Raw mango is generally green in color, changing to yellow or orange when fully ripe. The color change depends on the type of cultivar, and can also be an indicator of the quality of the fruit. There are several pigments, including chlorophylls, carotenoids, and flavonoids, that are responsible for the change in the color and metabolism of the mango fruit [58].

##### Chlorophylls

The green color of mango is due to the presence of chlorophyll. Two types of chlorophylls are found in mango fruit: chlorophyll a provides a blue-green color and chlorophyll b provides a yellow-green; these are present in a 3:1 ratio [59]. The chlorophyll content decreases as the fruit ripens, as the thylakoids start to collapse in the chloroplasts. Chlorophyll content decrease increases the carotenoid concentration in the pulp and peel of the fruit, and the color changes from green to golden yellow, red, or orange, depending on the variety. The chlorophyll content can be decreased by ethylene and by up-regulation of de novo synthesis of the chlorophyllase enzyme [60].

##### Carotenoids

Mango is one of the best sources of carotenoids; carotenoids are mostly responsible for the peel and flesh colors: yellow, orange, or red. Carotenoids, which exist in chromoplasts, are usually covered by chlorophyll and non-photosynthetic plant tissues [60]. Carotenoids present in the mango belong to two main groups: hydrocarbon carotenoids or carotenes (α-, β-, and γ-carotene) and xanthophylls or oxygenated derivatives (auraxanthin, antheraxanthin, neoxanthin, lutein, violaxanthin, cryptoxanthin and zeaxanthin). There are 25 different carotenoids that have been identified in the pulp and peel of mango. Among them, all-*trans*-β-carotene is the most abundant (around 60% of the total carotenoid content) followed by the all-*trans* and 9-*cis*-violaxanthin [25,35,56,61]. 

The carotenoid content usually changes depending on the fruit maturity stage and the local environment. Ellong et al. [27] reported four different varieties of carotenoids in the Bassignac variety, and the highest content (almost two-fold) was noted in fully ripened mango (4.138 mg/100 g) compared to unripe mango. According to the Nutrient Database of the USDA [35], the Tommy Atkins mango contain 0.64 mg β-carotene, 0.009 mg α-carotene, 0.01 mg β-cryptoxanthin and lutein, and 0.023 mg zeaxanthin per 100 g. Manthey and Perkins-Veazie [62] suggested that the variation in carotenoid content mainly depends on the type of cultivar, not on the location of production. 

#### 2.2.3. Flavonoids and Flavanols

The important phytochemicals with antioxidant and anti-inflammatory activities, including ascatechins, quercetin, anthocyanins, kaempferol, rhamnetin, and tannic acid, belong to a class of flavonoids. A high proportion of quercetin and its glycosides (46.6 mg/kg) and lower quantities of kaempferol, rhamnetin, myricetin, and fistien are present in the fresh mango pulp [42,56]. Among the condensed tannins and pro-anthocyanins, catechin presents in higher concentrations (1.72 ± 1.57 mg per 100 g fresh weight (FW)) compared to epicatechin (0.15 ± 0.0 mg per 100 g FW). In addition to the above pro-anthocyanin monomers, mango pulp contains dimers, trimers and tetra-hexamer compounds [58,63]. In addition to the above-mentioned condensed tannins, mango pulp contains hydrolysable tannins, gallotannins, and a wide array of their derivatives in small quantities (2 mg/100 g) [64]. The chemical structures of major bioactive molecules that are present in three parts of mango fruit are depicted in Figure 3.

#### 2.2.4. Phytosterols 

Mango pulp is known to have low amounts of lipids and fatty acids, but the mango seed is a rich source of lipids. Vilela et al. [65] analyzed the lipid profiles in twelve *M. indica* L. cultivars grown on Madeira Island using GC-MS, and reported similar quantity and quality compositions of free and glycosylated sterols (44.8–70.7%) and fatty acids (22.6–41.9%) in the total lipophilic component. Vilela et al. [65] further identified lower quantities of long-chain aliphatic alcohols and α-tocopherol. These findings revealed that the consumption of 100 g of fresh mango can significantly benefit health by providing 9.5–38.2 mg of phytosterols (free and glycosylated) and 0.7–3.9 mg of ω-3 and ω-6 fatty acids. 

## 3. Mango Peel

Mango fruit processing generates peel and kernel as the two main byproducts. Approximately 15–20% of the peel is not commercially used and causes pollution in landfills. However, mango peel consists of various valuable phytochemicals, including carotenoids, polyphenols, and other bioactive compounds, which have been reported to promote human health [66,67]. Owing to the high fiber content, mango peel has been used in a variety of food supplements to enhance their functional properties. Abbasi et al. [28] and Thokas et al. [68] have reported the presence of various bioactive constituents and dietary fiber in mango peel, which possesses antioxidant and free-radical-scavenging properties. 

### 3.1. Nutritional Composition

The mango peel composition mainly depends on the maturity stage, locality, variety, and climatic conditions in its production region. As shown in Table 1, mango peel contains a variety of macronutrients (total carbohydrates (20–30%), protein, amino acids, lipids, organic acids as well as dietary fiber) and micronutrients. Dietary fiber is an important functional nutrient and its concentration in different mango varieties, ranging between 16% and 28% soluble and 29% and 50% insoluble fiber [69]. The content of vitamin C ranges from 188–2570 µg/g, varying widely in different cultivars. Ajila et al. [69] reported 188–392 µg/g of vitamin C in both Badami and Raspuri varieties, and higher amounts in the ripened peel compared to the raw peel (Table 1). The presence of vitamin E (205–509 µg/g) in mango peel led to its use in the preparation of skin care products. The concentration of vitamin E is also higher in ripened mango peel than in raw mango peel (Table 1). The mango peel contains significantly higher levels than pulp of the following minerals: Ca > K > Mg > Na > Fe > Mn > Zn > Cu [70].

### 3.2. Phytochemical Composition

#### 3.2.1. Polyphenols

Mango peel has a higher polyphenol content than mango pulp at all growth stages of the fruit [55,71]. Many reports have been published on the polyphenolic content of the mango peel for various varieties available, and the variance mainly depends on the maturity stage, locality, variety, and climatic conditions in its production region. The polyphenol content in the peel ranges from 55–110 mg/gm dry weight, and higher levels are found in the ripe than in the unripe peel [72]. Two more important phytochemicals, quercetin 3-O-galactoside and mangiferin, are also present in the peel. It was estimated that mango peel has 1.69 g of mangiferin/kg dry weight, and this content is temperature-dependent. At high temperatures, mangiferin concentration decreases as its derivatives content increases [54]. Due to this transformation process, xanthones from the benzophenone derivative form, which is helpful in the formation of xanthone C-glycosides in the mango peel [64,67] (Table 1). For the Uba cultivar from Brazil, the polyphenol content is 6–8% of dry weight, and the amounts of flavonoids and xanthones in the peel are 4.6 and 7.3 times greater than in the pulp [55]. Of the six xanthone derivatives identified in mango peel, mangiferin (C2-β-D-glucopyranosyl-1,3,6,7-tetrahydroxyxanthone), a C-glucosyl xanthine, has many pharmacological activities. Mangiferin is a glucosyl xanthone with a characteristic isomeric form (mangiferin + isomangiferin + homo-mangiferin), and was found to be in higher concentrations in mango peel than in pulp and seed [42]. Mangiferin is a predominant bioactive compound in mango tree bark and is found in the fruits, roots, and leaves as well. The amount of mangiferin and its derivatives is higher in the peel than in the pulp (22.15 and 9.68 mg/100 g FW, respectively) and in Pica compared to Tommy Atkins mangoes (4.24 and 3.25 mg/100 g FW, respectively) [53]. Higher concentrations of mangiferin were reported from peels of Uba and Tommy Atkins mangoes (12.4 and 2.9 mg/kg dry weight, respectively), which are cultivated in Brazil [55]. 

In the peels of Tommy Atkins cultivar, 18 gallotannins were identified; the total concentration was 1.4 mg/g dry weight, expressed as gallic acid equivalents (GAE). The identified gallotannins are galloyl-glucose and quercetin derivatives. The major quercetins are quercetin-3-O-galactoside, quercetin-3-O-xyloside, quercetin-3-O-glucoside, quercetin-3-O-arabinofuranoside, and quercetin-3-O-arabinopyranoside [53,54,73]. The highest concentration of phenolic compounds (66.02 mg/100 g FW) is present in the Pica cultivar from Chile. Seven phenolic acid derivatives, three procyanidin dimers, and four xanthenes (homomangiferin, mangiferin, and mangiferin gallate), in a total of 13 compounds, were identified in the Pica peel [53]. 

#### 3.2.2. Carotenoids

Carotenoids are fat-soluble pigments that create the different fruit colors, such as yellow, orange, and red. Similar to pulp, mango peel contains high concentrations of carotenoids in the form of β-carotene, which provides vitamin A [55]. Unlike the pulp carotenoids, peel carotenoids have been less studied [74]. Ranganath et al. [75] analyzed the carotenoid compositions of different-colored mango peel at different phases of ripening, and identified eight carotenoids in 12 selected cultivars of various colors. They further identified content variation with respect to the color of the fruit. In all cultivars, β-carotene, cis-β-carotene, and violaxanthin isomers presented as the principal compounds. The highest content (31.18 µg/g FW) was observed in yellow-colored Arka Anmol, and the lowest content (0.74 µg/g) in Janardhan Pasand. The highest β-carotene concentration (13.01 µg/g FW) was reported in yellow-colored Arka Anmol, along with its presence in all the tested mango cultivars. The concentration of carotenoids usually increases during ripening and is high in the yellow-colored stage. 

Anthocyanins are water-soluble pigments; their presence provides red, blue, and purple colors to the fruits. These are presently used as biocolorants instead of synthetic colors [76]. Anthocyanins are known for their beneficial effects in the prevention of various diseases, including cancer, diabetes, and neuronal and cardiovascular diseases, thereby promoting human health [77,78,79,80]. Different ranges of anthocyanin concentrations have been reported by different authors. For example, total anthocyanin content ranges from 3600–5650 µg/g dry weight (DW) in the fully ripe stage and from 2030–3260 µg/g in the raw and unripe stages [72]. Berardini et al. [54] reported very low anthocyanin content (0.21–3.71 µg/g DW) in some red-colored mangoes (Tommy Atkins). Ranganath et al. [75] observed the highest anthocyanin content in red-colored mango peels (228.2 µg/100 g FW) compared to yellow- and green-colored mango peels. The major anthocyanins observed in different-colored peels of the various cultivars of mangoes include cyanidin, pelargonidin, delphinidin, malvidin, petunidin, and peonidin.

## 4. Mango Seed Kernel (MSK)

Consumption of fresh fruit by individuals and large-scale processing by the pulp industries, results in a significant quantity of mango seeds generated as a byproduct [22,81]. The MSK accounts for approximately 35–55% of the weight of the fresh fruit depending on the variety [82]. Landfilling of mango byproducts causes environmental problems because they do not decompose quickly. However, use of mango seed for the extraction of oil and phytochemicals may be economically profitable and environmentally safe. The mango seed is comprised of kernel (68%), shell (29%) and testa (3%) [83,84]. Although enough information is available on the nutritional composition of mango seed kernel, the composition varies mainly due to varietal and geographical differences. The composition of the MSK is unique and comparable with other oil-yielding seeds such as cocoa butter, shia, illipe, kokum, and sal butter [85,86,87]. 

### 4.1. Nutritional Composition

Like mango pulp, the seed is rich in nutrients, and has been used for developing various value-added products. The MSK contains high amounts of carbohydrates, protein, lipids, and several minerals [56,88,89].

#### 4.1.1. Carbohydrates

Mango seeds contain higher concentrations of carbohydrates: the 58–80% starch yields 21% of pure starch. The quality of mango seed starch is the same as tapioca starch [90] (Table 1). The quality and composition of carbohydrates mainly depend on the fruit variety and region of cultivation. Lakshminarayana et al. [91] analyzed 43 varieties cultivated all around India, and found a large variation in the concentrations of carbohydrates, proteins, and lipids. Sandhu and Lim [92] observed differences in the starch content of Chausa (75.6%) and Kuppi (80.0%) varieties. 

#### 4.1.2. Proteins and Amino Acids

The MSK contains 6–13% protein by DW and content, mainly depending on variety. Though the protein content is low in the seed, it is nutritious protein because of its essential amino acid composition, including leucine (6.9 g), isoleucine (4.4 g), methionine (1.2 g), phenylalanine (3.4 g), lysine (4.3 g), threonine (3.4 g), tyrosine (2.7 g), and valine (5.8 g) per 100 g protein [93]. These essential amino acids, except methionine, occur in higher levels in the MSK than the Food and Agriculture Organization (FAO)-referenced protein: the essential amino acids content in MSK is 70% higher compared to the standard proteins [94] (Table 1). The indigestibility and toxic nature of MSK flour is due to the presence of higher concentrations of tannins. Nevertheless, the protein present in MSK is high quality due to its high essential amino acid content and protein quality index [93,95].

#### 4.1.3. Lipids and Fatty Acids

Lipids are highly nutritious, have high energy values, and are well-known for their functional properties. Mango seed kernel, containing 8.15% to 13.16% of oil, is an appealing lipid source because of its nutritive and health beneficial properties [96]. The physical and chemical properties of the lipids and fatty acids present in the MSK are listed in Table 2. The MSK fat characteristics are comparable with those of vegetable butter [86], with a predominant presence of stearic and oleic acids. In some varieties, the unsaturated fatty acid content, especially linoleic acid, is twice and even three times higher than that of saturated fatty acids [94]. The high stearic and oleic acid contents in the total fatty acid profile of the MSK is attributed to its higher stability than polyunsaturated fatty acids (PUFAs)-containing oils. Hence, the MSK oil is suitable and safe for daily cooking [85,96]. Furthermore, the antioxidant potency of 1% mango seed crude oil extract is comparable with that of 200 ppm butylated hydroxytoluene (BHT) [97]. The good quality of edible oil in MSK is comparable to soybean and cotton seed oil. The total phenolic content and induction period of MSK oil is greater than several commercial vegetable oils [98]. The yield of MSK oil can be increased by increasing temperature, time and volume of extraction solvent [84]. However, the safety and suitability of MSK oil is depends on the extraction procedure and quality of the MSK.

The fatty acids obtained from the MSK have many similarities with cocoa butter in terms of properties, such as solid fat content, triglycerides, crystallization, and melting point; therefore, it used to replace cocoa butter. Premium-grade mango seed fat and oils (extracted by modern technologies like supercritical fluids) have premium characteristics and, along with palm stearin, can be used in the preparation of temperature-resistant chocolates in tropical countries [87].

#### 4.1.4. Minerals and Vitamins

The MSK contains high levels of minerals, specifically potassium (368 mg/100 g), calcium (170 mg/100 g), and phosphorus and magnesium (210 mg/100 g). The MSK also contains vitamins C and E (antioxidant vitamins) and other essential vitamins, including K, B1, B2, B3, B5, B6, B9, and B12, in various concentrations ranging from 0.1 to 1 mg per 100 g. In addition, about 15 IU of vitamin A can be found in the MSK. The MSK is one of the rich sources of vitamin B12 (0.12 mg/100 g), which is higher than the recommended daily intake of the vitamin (2–3 µg); therefore, MSK can be used to prevent vitamin B12 deficiency in vegetarians (Table 1) [70,102]. 

### 4.2. Phytochemical Composition

Similar to the pulp and the peel, the MSK is also considered a prospective source of polyphenols with potent antioxidant activity. Ahmed et al. [93] estimated that MSK extract contains about 112 mg GAE/100 g of total polyphenols, and the identified constituents were tannins (20.7 mg), gallic acid (6.0 mg), coumarins (12.6 mg), caffeic acid (7.7 mg), vanillin (20.2 mg), mangiferin (4.2 mg), ferulic acid (10.4 mg), cinnamic acid (11.2 mg), and unidentified compounds (7.1 mg/100 g) (Table 1). Soong and Barlow [103] found that the antioxidant activity of the MSK is greater than that of other fruit seeds, namely jackfruit, longan, and avocado. These reports suggest that the polyphenolic content of the MSK contributes to its potent antioxidant activity. Approximately 75% of the total tannins are in hydrolyzable form, which need to be processed before using in food and feed preparations to reduce their toxic effect in vivo. A large difference in the quantity of gallic acid (23 to 838 mg/100 g) and ellagic acid (3 to 156 mg/100 g GAE) was reported in the MSK. The high compositional variation was attributed to the differences in extraction methods (Table 1) [103]. Use of ellagitannins from natural sources was found to be more effective than using purified ellagic acid. The total phenolic content (TPC) of mango seed oil ranges from 9–10 TPC mg/g. Kittiphoom and Sutasinee [104] studied the composition of various phenolic compounds in MSK using the HPLC technique. For the first time, they reported hesperidin as the major compound. The MSK was further characterized using the number of functionally important phenolic compounds, phenolic acids, and antioxidant minerals (selenium, zinc, copper, manganese, and potassium). Among them, quercetin, mangiferin, isomangiferin, homomangiferin, kaempferol, and anthocyanins were found to be the phenolic compounds; gallic acid, caffeic, protocatechuic, coumaric, ferulic, and ellagic acids were reported to be the phenolic acids [88]. 

In the MSK, the total flavonoid content was estimated as about 3325 ± 120 mg catechin equivalent (CE)/100 g seed [105]. The reported flavonoids content was higher than those reported in previous investigations, at about 10–1170 mg/100 g [55,69,106,107]. With the rich presence of functional compounds, the MSK shows potential for the preparation of functional foods with health benefits.

## 5. Therapeutic and Health-Promoting Effects of Mango Fruit

In India, the whole mango tree, including the stem, bark, leaves, flowers, and fruit, has been widely used as an ancient traditional medicine to treat various diseases and discomforts. All parts of the mango tree contain essential bioactive compounds, such as mangiferin, quercetin, catechins, and kaempferol. In recent years, many researchers have explored the ethnopharmacological and pharmacological efficacies of the various bioactive constituents of mango fruit. This evidence emphasizes the importance of mango byproducts in the treatment of various chronic diseases, including diabetes, cancer, asthma, hypertension, and hemorrhage in the lungs and intestine, having higher efficiency and fewer side effects [56]. Different parts of the mango offer different benefits, such as anti-inflammatory, antioxidant, anticancer, anti-diabetic, antimicrobial, anti-hyperlipemic, and immunomodulatory activities (Figure 4) [56,69,108]. 

So far, about 500 research articles have been published on the presence, structure, chemical synthesis, and therapeutic properties of mangiferin. The majority of these studies are from India and Bangladesh. Countries like Brazil, Nigeria, and Iran also contributed significantly to reporting the therapeutic effects of mango contents. Mangiferin has been shown to restore the learning and memory impairments and to decrease hippocampal injury in a mouse model [109]. A recent study demonstrated that mangiferin potently inhibits the progression of human epithelial ovarian cancer [14]. A human study revealed that supplementation with mangiferin combined with luteolin enhances the exercise spring performance and brain oxygenation during sprint exercise in physically active men [110]. Mango intake (200–400 g of pulp for 8 weeks) has been shown to decrease inflammatory biomarkers and improve the intestinal microbiota in patients with inflammatory bowel disease [111]. Another report documented that the intake of 85 g of mangos (16 weeks) reduced facial wrinkles in fair-skinned postmenopausal women [112]. In addition to these, other health-benefiting properties of the mango pulp, peel, and kernel are listed in Table 3.

## 6. Conclusions

Mango is the main fruit produced in India, accounting for almost 55% of the global production [3]. Mango is one of the best sources of nutrient, such as carbohydrates, proteins, and fatty acids. Various parts of mango contain several bioactive phytochemical compounds, namely polyphenols, carotenoids, flavonoids, tannins, and vitamins. The amounts of bioactive and phytochemicals are different in the epicarp, pericarp, and mesocarp. In the recent years, mango cultivation and research has gained popularity due to their potent antioxidant and anti-inflammatory properties. Mango pulp is processed into a number of high-value products, like juice, puree, and jam, with the peel and kernel as byproducts. Mango peel is known to contain pectin, dietary fiber, vitamins, carotenoids, and phenolic compounds, having health-promoting effects. Mango seed contains notable amounts of starch, essential amino acids, and oil. Mango seed oil is rich in oleic and stearic acids, and contains different phytochemicals. Mango seed has been used in the production of mango butter and seed flour, which are used in functional foods. The proper use of mango peel and seed (raw materials) in food and feed preparation not only improves the economy, but also reduces the environmental impacts. The bioactive compounds found in the three parts (pulp, peel and seed) of *M. indica*, such as mangiferin, gallic acid, catechin, quercetin, β-carotene, shikimic acid, and kaempferol, have been reported to have antioxidant effects. These compounds are also well-known for their anticancer, anti-diabetic, anti-inflammatory, skin-protecting, neuron-protecting, antimicrobial, and anti-aging effects. However, more pharmacokinetics, pharmacodynamics, and clinical trials are required to assess the toxicity or the safe dose of these molecules. Nevertheless, mango is considered safe and even beneficial for cellular functioning, and regular consumption of fresh mango fruit and/or use of its byproducts could promote overall health. 

## Figures and Tables

**Figure 1 ijerph-18-00741-f001:**
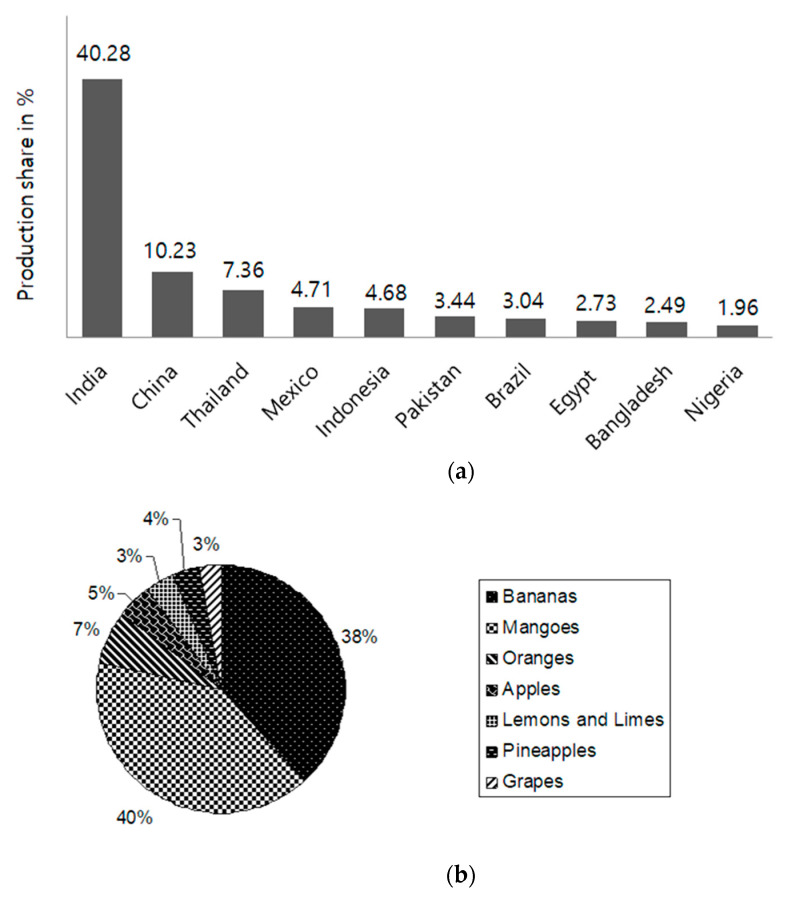
(**a**) Production share (%) of the top ten mango-producing countries in 2019–2020. (**b**) Area (thousand ha) covered under mango crop in India (2019–2020). Sources: National Mango Database 2020 [1]; APEDA 2020 [3].

**Figure 2 ijerph-18-00741-f002:**
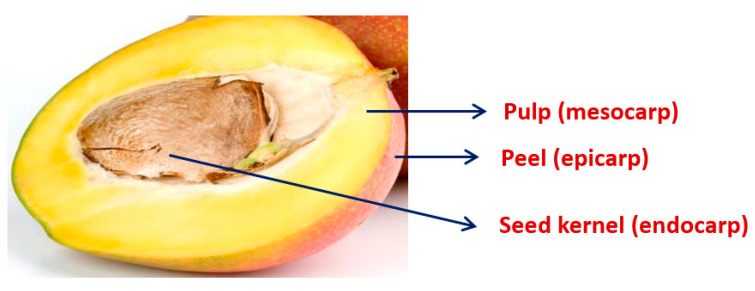
Major parts of mango fruit.

**Figure 3 ijerph-18-00741-f003:**
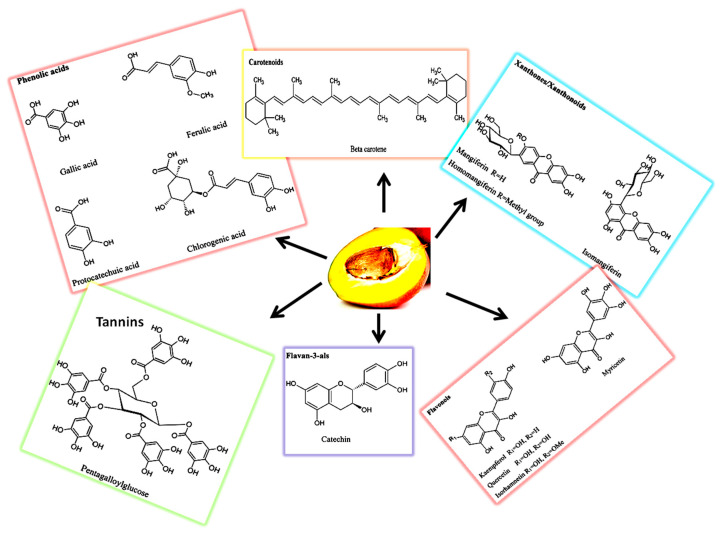
Major bioactive compounds and their chemical structures in the three parts of mango fruit.

**Figure 4 ijerph-18-00741-f004:**
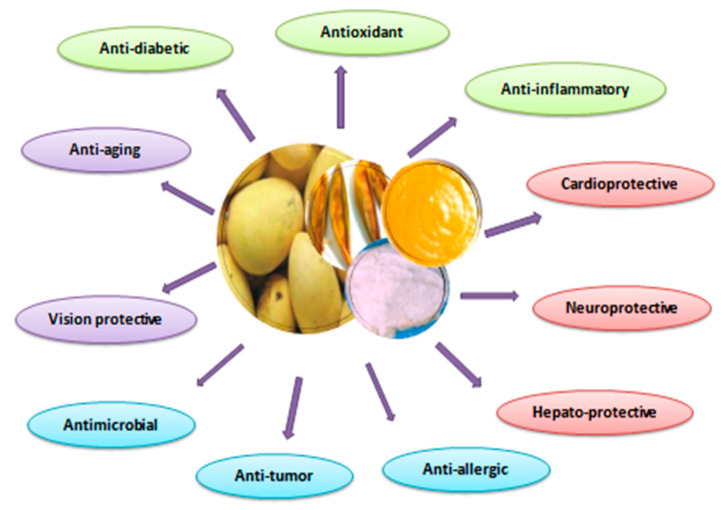
Potential health benefits of bioactive compounds in mango fruit.

**Table 2 ijerph-18-00741-t002:** Physical and chemical characteristics and composition of mango seed oil.

Characteristic/Composition	Content
Oil (%)	11.5
Free fatty acid (FFA%)	0.22
Moisture (%)	0.21
Iodine Value	54.6
Refractive Index	1.457
Melting Point (°C)	35.2
Saponification	193
Unsaponifiable Matter (%)	1.68
Peroxide Value (meq O_2_/kg)	0.65
Color	R2.8 + 30Y
C16:0	7.43
C18:0	37.5
C18:1	45.59
C18:2	5.48
C18:3	0.40
C20:0	2.48
C22:0	0.45
C24:0	0.40
Campesterol	0.07%
Stigma Sterol	10.66
β-Sitosterol	58.63
Δ5-Avenasterol	10.19
Δ7-Stigmasterol	4.34
Δ7-Avenasterol	19.10

Sources: Abdel-Razik et al. [99] Azeem et al. [100], and Dhara et al. [101].

**Table 3 ijerph-18-00741-t003:** Biological activities of phytochemicals isolated from *M. indica*.

Biochemical Property and Name of Part Used	Name of the Compound	Reference
**1. Antioxidant Activity**		
Peel	3,4-Dihydroxybenzoic acid (protocatechuic acid)	[63]
Fruit pulp	Kaempferol	[73,113]
Fruit pulp	Linalool	[114]
Fruit pulp	Mangiferin	[115]
Fruit pulp	Quercetin	[116,117,118]
Fruit pulp	β-Carotene	[119]
Peel	β-Carotene	[120]
**2. Anti-Inflammatory Activity**		
Peel	5-(11*Z*-Heptadecenyl)-resorcinol and	[121]
	5-(8,*Z*,11,*Z*-Heptadecadienyl)-resorcinol	
Fruit pulp	Kaempferol	[55,73,113]
Fruit pulp	Mangiferin	[115,122,123]
Fruit pulp	Shikimic acid	[124]
**3. Antimicrobial Activity**		
Fruit pulp	Kaempferol	[55,73,113]
Fruit pulp	Mangiferin	[115,122,123]
Fruit pulp	Quercetin	[116,117,118]
Peel	3,4-Dihydroxybenzoic acid	[63]
	(protocatechuic acid)	
**4. Anti-Diabetic and Anti-Obesity Activity**		
Fruit pulp	Kaempferol	[55,73,113]
Peel	3,4-Dihydroxybenzoic acid (protocatechuic acid)	[63]
Fruit pulp	Mangiferin	[115,122,123]
Fruit pulp	β-Carotene	[119]
Fruit pulp	Quercetin	[116,117,118]
**5. Cytotoxic and Apoptotic Activity**		
Fruit pulp	Kaempferol	[55,73,113]
Fruit pulp	Linalool	[114]
Fruit pulp	Mangiferin	[115,122,123]
Fruit pulp	Quercetin	[116,117,118]
Fruit pulp	β-Carotene	[119]
Peel	3,4-Dihydroxybenzoic acid	[63]
	(protocatechuic acid)	
Seed	Mangiferin	[15]
Seed	Gallic acid	[103]
**6. Neuroprotective Activity**		
Fruit pulp	Linalool	[125]
Fruit pulp	Mangiferin	[115,122,123]
**7. Cardio-Protective**		
Fruit pulp	Mangiferin	[115,122,123]
**8. Anticoagulant/Antithrombotic**		
Fruit pulp	Shikimic acid	[126]
**9. Blood-Pressure-Lowering Activity**		
Fruit pulp	Quercetin	[116,117,118]
**10. Gastro-Protective**		
Seed	Mangiferin	[103]

## Data Availability

Data sharing is not applicable.

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
