# Peer review of "Nutritional Composition and Bioactive Compounds in Three Different Parts of Mango Fruit"

_ijerph, 2021, doi:10.3390/ijerph18020741_

Round 1

Reviewer 1 Report

Dear author,

Of the review “Nutritional composition and bioactive compounds in 2 three different parts of mango fruit” I appreciate two aspect:

The first one is the description of three different constituent of the whole fruit (pulp, peel and kernel) and the second one is the dissertation about the relationship between the cultivars and the content of the different phytochemicals. Usually, these data are precious for researchers working on the same topics.

However, I’ve found different negative aspects. First, the language that need a deep revision. Moreover, in few parts, that I listed in the bottom section, there were not any correspondences between the text and the indicated table.

Despite these considerations, in my opinion the review should be considered for publications after major revision.

Lines 97-100: these lines concern peel and kernel composition, but the paragraph is about the Mango pulp. These lines should be removed because they cause an interruption of the paragraph.

Line 120: The table number should be 1?

Table 1: which is the source of these data? The author should better specify.

Line 140: “The usually occurring amino acids are listed in Table 1”. Please, verify if the table content is correct. Probably amino acids were not included.

Line 380: “the seed is also contains rich quantities of nutrientsthe sentence should be changed in “the seed also contains rich quantities of nutrients”.

Line 386: “The quality and composition is mainly depends on the variety and region of cultivation”. The sentence should be corrected, because this form seems to be grammatically incorrect.

398: Indigestibility and toxic nature of MSK flour at higher levels ii because of the high concentration protein of tannins: The sentence should be revised.

Line 410: “MSK oil is good and safe for daily cooking”. In my opinion, the safety of this oil depends not only on its composition but also on the extractive methods. If the extraction is carried out with “solvent-free” methods, the product can be considered save and healthy but in case of extraction with solvent, a daily intake can be considered quite “unhealthy”. I think that the author should added a short digression about the different available methods of extraction of oil from mango kernel and the characteristic of the product (especially yield).  

Best regards

Author Response

Responses to Reviewers’ Comments: ijerph-1029417

Reviewer 1

Dear author,

Of the review “Nutritional composition and bioactive compounds in 2 three different parts of mango fruit” I appreciate two aspect:

The first one is the description of three different constituent of the whole fruit (pulp, peel and kernel) and the second one is the dissertation about the relationship between the cultivars and the content of the different phytochemicals. Usually, these data are precious for researchers working on the same topics.

However, I’ve found different negative aspects. First, the language that need a deep revision. Moreover, in few parts, that I listed in the bottom section, there were not any correspondences between the text and the indicated table.

Despite these considerations, in my opinion the review should be considered for publications after major revision.

Dear Reviewer,

We are highly thankful to the Reviewer for providing the valuable comments on our manuscript. We agree with the comments and these comments helped us to improve the quality of our manuscript. According to the comments, we have extensively revised our manuscript carefully, and all corrections were marked in red color. The revised manuscript has been edited by MDPI English editing service (English editing ID: english-25650). Our point-by-point responses to each comment were given below: 

Comment 1: Lines 97-100: these lines concern peel and kernel composition, but the paragraph is about the Mango pulp. These lines should be removed because they cause an interruption of the paragraph.

Response: We agree with the Reviewer’s opinion. As suggested the lines has been removed.

Comment 2: Line 120: The table number should be 1?

Response: We are apologizing for the typo. Now the Table number has been corrected as ‘Table 1’ in the revised manuscript, Page 4, Line 130.

Comment 3: Table 1: which is the source of these data? The author should better specify.

Response: We thank reviewer for this comment. Now the source of the data presented in Table has been given at the bottom of the Table 1. 

Comment 4: Line 140: “The usually occurring amino acids are listed in Table 1”. Please, verify if the table content is correct. Probably amino acids were not included.

Response: We are apologizing for the typo. The amino acids were not listed in the Table 1. However, amino acids description was provided in Page 4, Lines 150-152 and also in Page 11 Line 404-406.

Comment 5: Line 380: “the seed is also contains rich quantities of nutrients” the sentence should be changed in “the seed also contains rich quantities of nutrients”.

Response: We appreciate Reviewer’s suggestion. Now the sentence has been changed as “Like mango pulp, the seed is rich in nutrients, and has been used for developing various value-added products” in the manuscript, Page 11, Line 393-395.

Comment 6: Line 386: “The quality and composition is mainly depends on the variety and region of cultivation”. The sentence should be corrected, because this form seems to be grammatically incorrect.

Response: As suggested, now the sentence has been revised for the clarity in the revised manuscript, Page 11, Line 398-399. The revised sentence read as “The quality and composition of carbohydrates mainly depend on the fruit variety and region of cultivation”

Comment 7: 398: Indigestibility and toxic nature of MSK flour at higher levels ii because of the high concentration protein of tannins: The sentence should be revised.

Response: We thank Reviewer for bringing out the typo. Now we have revised the sentence as “The indigestibility and toxic nature of MSK flour is due to the presence of higher concentrations of tannins”, in the revised manuscript, Lines 410-411. 

Comment 8: Line 410: “MSK oil is good and safe for daily cooking”. In my opinion, the safety of this oil depends not only on its composition but also on the extractive methods. If the extraction is carried out with “solvent-free” methods, the product can be considered save and healthy but in case of extraction with solvent, a daily intake can be considered quite “unhealthy”. I think that the author should added a short digression about the different available methods of extraction of oil from mango kernel and the characteristic of the product (especially yield).  

Response: Authors are thankful to the Reviewer for this interesting comment. We do agree with the Reviewer’s opinion that the safety of oil is depends on extractions methods as well. Within the scope of our review, we have included suitable information about MSK oil in the revised manuscript, page 11, Lines 424-428. 

Reviewer 2 Report

-Lines 81-83 need a reference. Please avoid using too general claims. 

-Table 1 needs references for all the presented information.

-Line 153: citric acid. Also, in the text sometimes is mangos and sometimes mangoes 

-Line 195-196 needs to be rephrased.

-English language needs to be improved/corrected in some parts.

Author Response

Responses to Reviewers’ Comments: ijerph-1029417

Reviewer 2

We express our sincere thanks to the Reviewer for his/her evaluation and positive comments on our manuscript. We agree with the comments, and we have revised the manuscript accordingly. All the corrections were marked in red color and detailed response to each comment was given below.

Comment 1: Lines 81-83 need a reference. Please avoid using too general claims. 

Response: Authors are thankful to the Reviewer for this comment. As suggested we have revised the sentence and appropriate references were added to support the statement. The revised sentence read as “High concentrations of β-carotene and other phytochemicals in mangoes can prevent leukemia and progression of prostate, breast, and colon cancers”, Lines 92-94.

Comment 2: Table 1 needs references for all the presented information.

Response: Apologizing for not providing the source for Table 1. Now we have included the source references for all information presented in Table 1 (Page 6, Lines: 181-182) 

Comment 3: Line 153: citric acid. Also, in the text sometimes is mangos and sometimes mangoes.

Response: We noticed the typo in Line 192 (in the revised manuscript), apologizing for that. All typos have checked and corrected in the whole manuscript. Usage of mango or mangoes also checked and corrected accordingly. The revised manuscript has been edited by MDPI English editing service (english-25650).

Comment 4: Line 195-196 needs to be rephrased.

Response: As suggested the Lines 208-209 (in the revised manuscript) have been rephrased as “Phenolic acids are the important secondary metabolites that help to protect against various diseases and play pivotal role in management of human health” (Page 7).

Comment 5: English language needs to be improved/corrected in some parts.

Response: As suggested, English language has been improved in the manuscript. We have used MDPI English editing service (English editing ID: english-25650) and certificate was appended below.

Reviewer 3 Report

Strengths 

  • The authors used all three different parts (pulp, peel, and seed kernel) of mango to provide information on nutrition, composition, and bioactive compounds.
  • They also presented proper tables, nutritional and functional phytochemical composition of mango pulp, peel, and kernel.
  • Biological activities of phytochemical isolated from M. India was tabulated nicely.

Weaknesses 

  • Several grammar and typing errors: lines 33, 153 and 283
  • Abbreviations need to be added
  • beta-cryptoxanthin is missing in line 233
  • please stick with one nomenclature for carotenoids - use either symbols or roman letters and stick with that throughout the manuscript.

Author Response

Responses to Reviewers’ Comments: ijerph-1029417

Reviewer 3

Strengths 

  • The authors used all three different parts (pulp, peel, and seed kernel) of mango to provide information on nutrition, composition, and bioactive compounds.
  • They also presented proper tables, nutritional and functional phytochemical composition of mango pulp, peel, and kernel.
  • Biological activities of phytochemical isolated from India was tabulated nicely.

Dear Reviewer,

We are highly thankful the Reviewer for the keen evaluation and useful comments on our manuscript. We have answered all the comments and made appropriate corrections in the revised manuscript. The detailed point-by-point response to each comment was given below. All the corrections in the revised manuscript were marked in ‘red’ color.   

Comment 1: Several grammar and typing errors: lines 33, 153 and 283

Response: All the grammatical and typing errors were carefully corrected in the revised manuscript. Along with the corrections in Line 34-34, 192 and 296 (in the revised manuscript), the whole manuscript has been edited by MDPI English editing service (English editing ID: english-25650).

Comment 2: Abbreviations need to be added

Response: Authors are thankful to the Reviewer for this comment. As suggested all abbreviations were fully defined wherever applicable in the manuscript, including Abstract, Text and Tables.

Comment 3: beta-cryptoxanthin is missing in line 233

Response: We appreciate Reviewer for this suggestion. Now we have included the cryptoxanthin in the revised manuscript, Page 7, Line 247.

Comment 4: please stick with one nomenclature for carotenoids - use either symbols or roman letters and stick with that throughout the manuscript.

Response: We thank Reviewer for pointing out the inconsistency. We used the symbol to express the carotenoids, and same style was followed throughout the manuscript.

Reviewer 4 Report

Review for MDPI IJERPH-1029417

Nutritional composition and bioactive compounds in three different parts of mango fruit

Overall:  Very good review of literature looking at the benefits of biological compounds of different parts of the mango fruit as they apply to health as well as economy and environmental benefits.  The Introduction could use a lot of grammatical editing.  The rest of the paper is relatively well-written but there are a few places where additional sources and clarification would be beneficial.  See below for detailed comments. 

Abstract

 Line 14 - The term “amazing” is subjective and should be replaced with something that has support in research.  I suggest replacing it with either another objective adjective or something suggest popularity of taste that can be supported through a study. 

Abstract Overall – include specific quantitative comparisons between mangos and other fruits (like avocados) to support the argument that these unused parts of the mango have equal beneficial value when compared to other popular fruits with pits/large seeds.

Introduction

Overall – Major grammatical edits are required with some additional sources needed.

Line 33 – delete “is” from “mango is belongs to…” as well as changing “Which is having” to “which contains a number of…”

Line 34 – regarding delicious taste, aroma, and nutrition, these should include sources and references pertaining to the popularity and the sentence should focus on the regional, national, or international popularity of the fruit based on export or consumption data

Line 38 – “occupied first place” should be edited to something closer to “remains the top exporter”

Line 39 – “state is a great place for the vast..” should be reworded to something closer to “is ideal for the cultivation of…”

Line 41 – change “cultivating” to cultivated

Line 42 – change “cultivating” to cultivated

Line 43 – change “which enabled the highest ranking” to “the highest level of production in the entire country”

Line 46 – change “while” to “with”

Line 47 – delete “with the major production accounts” and change to “accounting for 80% of the total…”

Figure 1 – what are the sources for these graphics?

Line 66 – change “and occupied third position” to “as the third largest agricultural product

Line 70 – change “cultivating” to “cultivated”

Lines 73-74 – a source is needed for this

Line 75 – grammatical edits are needed in this sentence to account for plurals and clarification

Line 76 – reword sentence for clarity “During the maturation of mangos…”

Lines 81-82 – sources needed for the link of elements to mangos to cancer

Line 86 – add “is” in front of “used”

Line 89 – source needed for annual growth

Line 90 – editing needed to clarify “profusely considering”

Line 92 – Do you mean “or” instead of “of”?

Line 93 – “byproducts” instead of “by products” and sources are needed for this

Line 95 – Edit sentence to read more clearly to indicate that the residue causes environmental pollution (and add sources)

Line 96 – The sentence states that the residue is used to feed cattle but the previous sentence states that it is pollution.  Please clarify.

Line 97 – change “as” to “is”

Line 100 – add sources to the end of the sentence about the properties of mangos

Line 101 – change “as nutrients could” to “as nutrients that could serve as” or “as nutrients that could”

Line 103 – add “that” between “compounds” and “provide”

Nutritional Composition – add references/citations for mango composition

Line 153 – no comma needed after “organic acids”

Line 283 – a space is needed between “reported” and “similar”

Line 304-305 – Is there a source/reference for the information on vitamin C?

Line 371 – When discussing the percentage, please specify the metrics associated with the percentage (e.g. weight, total number, etc)

Line 472 – Is there a source for the percentage of global production?

Author Response

Responses to Reviewers’ Comments: ijerph-1029417

Reviewer 4

Review for MDPI IJERPH-1029417

Nutritional composition and bioactive compounds in three different parts of mango fruit

Overall:  Very good review of literature looking at the benefits of biological compounds of different parts of the mango fruit as they apply to health as well as economy and environmental benefits.  The Introduction could use a lot of grammatical editing.  The rest of the paper is relatively well-written but there are a few places where additional sources and clarification would be beneficial.  See below for detailed comments. 

We are highly thankful to the Reviewer for his/her encouraging comments and detailed suggestions to revise our manuscript. All the comments are important that gave us an opportunity to strengthen our manuscript. We have answered all the comments and made appropriate changes in the revised manuscript. The detailed point-by-point response to each comment was given below. All the corrections in the revised manuscript were marked in ‘red’ color.   

Comment: Line 14 - The term “amazing” is subjective and should be replaced with something that has support in research.  I suggest replacing it with either another objective adjective or something suggest popularity of taste that can be supported through a study.

Response: Authors are thankful to the Reviewer for this comment. As suggested, we have replaced the term ‘amazing’ with “attractive” in the Abstract line 14.  The correction was confirmed by the MDPI English editing service (English editing ID: english-25650).

Comment: Abstract Overall – include specific quantitative comparisons between mangos and other fruits (like avocados) to support the argument that these unused parts of the mango have equal beneficial value when compared to other popular fruits with pits/large seeds.

Response: We agree with Reviewer that quantitative comparison of phytochemicals between mango and other tropical fruits could offer broad spectrum of knowledge. However, this will enable readers to focus on multiple dimensions of information on fruits other than mango, and will be out of our scope. The scope of our review is to compare the phytochemicals, nutritional values and therapeutic benefits of three parts of mango fruit. In view of this, we have provided relevant quantitative data for mango pulp, peel and seed kernel in Table 1, and explained under respective subheadings. Due to word limit (total 200 words) it is hard to provide the quantitative data in the Abstract. 

Comment: Line 33 – delete “is” from “mango is belongs to…” as well as changing “Which is having” to “which contains a number of…”

Response: Authors are thankful to the Reviewer for this meaningful comment. Now we have deleted ‘is’ and also changed the following words to read as “Mango belongs to the Anacardiaceae family, which contains a number of deadly poisonous plants”, Lines 32-33.

Comment: Line 34 – regarding delicious taste, aroma, and nutrition, these should include sources and references pertaining to the popularity and the sentence should focus on the regional, national, or international popularity of the fruit based on export or consumption data.

Response: We have changed the sentences as suggested in Page 1. We would like to bring to Reviewer’s notice that this is just an introductory statement, and all supporting references for taste, nutritional values and production have been given in the later parts of the manuscript, wherever applicable.

Comment: Line 38 – “occupied first place” should be edited to something closer to “remains the top exporter”

Response: We appreciate Reviewer for this suggestion. As suggested, we have revised the sentences from Line 35 to 37. 

Comment: Line 39 – “state is a great place for the vast..” should be reworded to something closer to “is ideal for the cultivation of…”

Response: As per the Reviewer suggestion, now we have modified the sentence from Line 37 to 39 that read as “Due to the optimal geographical and climatic conditions, the state of Andhra Pradesh is ideal for the fruit and vegetable cultivation”

Comment: Line 41 – change “cultivating” to cultivated

Response: The word ‘cultivating’ has been changed to ‘cultivated’ in the manuscript, Line 40.

Comment: Line 42 – change “cultivating” to cultivated

Response: The word ‘cultivating’ has been changed to ‘cultivated’ in the manuscript, Line 40.

Comment: Line 43 – change “which enabled the highest ranking” to “the highest level of production in the entire country”

Response: As suggested, the sentence has been changed to “the highest level of production in the entire country” in the revised manuscript, Lines 40-42.

Comment: Line 46 – change “while” to “with”

Response: The whole sentence (lines 44-46) has been revised to give appropriate meaning.

Comment: Line 47 – delete “with the major production accounts” and change to “accounting for 80% of the total…”

Response: As suggested, now the whole sentence from line 44 to 46 has been changed to “Mango is cultivated in 85 countries worldwide; Asian countries, including India, China, Thailand, and Indonesia provide for 80% of the total world production”

Comment: Figure 1 – what are the sources for these graphics?

Response: We thank Reviewer for this comment. Now the sources were incorporated for Figure 1 (Lines 62-63).

Comment: Line 66 – change “and occupied third position” to “as the third largest agricultural product

Response: As per Reviewer’s suggestion, we have revised the sentence as “Economically, mango is important, being the third-largest agricultural product in India” Line 64.

Comment: Line 70 – change “cultivating” to “cultivated”

Response: For better understanding, the term ‘cultivating’ has been changed to “cultivars” in the revised manuscript, Line 67.

Comment: Lines 73-74 – a source is needed for this

Response: The supporting reference is now added to the statement, Lines 85-87 in the revised manuscript.

Comment: Line 75 – grammatical edits are needed in this sentence to account for plurals and clarification.

Response: Now the sentence (86-87) was clearly rewritten with correct meaning, and also edited by MDPI English editing service.

Comment: Line 76 – reword sentence for clarity “During the maturation of mangos…”

Response: The whole sentence was restructured accordingly in the revised manuscript, Lines 87 to 90.

Comment: Lines 81-82 – sources needed for the link of elements to mangos to cancer.

Response: As suggested, supporting references for the sentence were included in the revised manuscript Line 94. 

Comment: Line 86 – add “is” in front of “used”

Response: As suggested “is’ has been added before ‘used’, Line 110 in the revised manuscript.

Comment: Line 89 – source needed for annual growth

Response: The source for the statement has been included (Line 113).

Comment: Line 90 – editing needed to clarify “profusely considering”

Response: We have edited the sentence and changed ‘profusely considering’ to “used extensively” to get the clear meaning (Page 3, Line 114).

Comment: Line 92 – Do you mean “or” instead of “of”?

Response: We have revised the sentence and deleted the redundancy words, Page 3 Lines 116-117.

Comment: Line 93 – “byproducts” instead of “by products” and sources are needed for this.

Response: The word has been corrected as “byproducts” and appropriate references have been cited as suggested, Line 118.

Comment: Line 95 – Edit sentence to read more clearly to indicate that the residue causes environmental pollution (and add sources).

Response: The sentence has been revised for the clarity and source references were also included the revised manuscript, Line 120.

Comment: Line 96 – The sentence states that the residue is used to feed cattle but the previous sentence states that it is pollution.  Please clarify.

Response: Mango waste is used as cattle feed in some areas. However, bulk quantities of mango waste from factories are not easy to handle. As a result accumulation of mango waster in open fields can cause pollution. This statement has been revised for the clarity.

Comment: Line 97 – change “as” to “is”

Response: In response to answer another Reviewer’s comment, we have removed this sentence, as it is describing about the peel and seed kernel under the mango pulp subheading.

Comment: Line 100 – add sources to the end of the sentence about the properties of mangos

Response: Similar to the above comment, we have removed this sentence as it describes about pulp ad kernel under pulp subsection.

Comment: Line 101 – change “as nutrients could” to “as nutrients that could serve as” or “as nutrients that could”

Response: Authors are thankful to the Reviewer for this suggestion. Now we have completely revised the sentence for the clarity in the revised manuscript, lines 121-123.

Comment: Line 103 – add “that” between “compounds” and “provide”

Response: We appreciate Reviewer for his/her keen evaluation. For better understanding, we have restructured and revised the mentioned sentence, lines 123-125. 

Comment: Nutritional Composition – add references/citations for mango composition.

Response: We thank Reviewer for pointing out the missing references. Now we have added suitable references for nutritional composition in mango pulp.

Comment: Line 153 – no comma needed after “organic acids”

Response: The comma after organic acids was removed in the revised version, Line 192.

Comment: Line 283 – a space is needed between “reported” and “similar”

Response: As suggested, the space between ‘reported’ and ‘similar’ was incorporated, Line 296 (Page 9).

Comment: Line 304-305 – Is there a source/reference for the information on vitamin C?

Response: We would like to update Reviewer that information on vitamin C is our own statement, based on the existing reports. In this statement, we have explained the range of vitamin C in different cultivars (Lines 317-318). Subsequently, the exact quantity of vitamin C in different varieties was provided with suitable references, Lines 318-323.

Reference: Line 371 – When discussing the percentage, please specify the metrics associated with the percentage (e.g. weight, total number, etc).

Response: We are thankful to the Reviewer for this useful comment. Now the sentence has been revised as “The MSK accounts for approximately 35–55% of the weight of the fresh fruit depending on the variety” in the revised manuscript, Line 383-384. We have further checked and corrected similar errors in the whole manuscript.

Comment: Line 472 – Is there a source for the percentage of global production?

Response: Yes, the reference to support the global production was included in the revised manuscript, Line 508.

Round 2

Reviewer 1 Report

Dear author,

the new version of the review "Nutritional composition and bioactive compounds in three different parts of mango fruit" in my opionion resulted to be improved as concern the aspect highlighted in my comments.

I suggest to accept the review for publication.

Best reguards